# Comprehensive Analysis of Different Coating Materials on the POM Substrate

**DOI:** 10.3390/ma16124365

**Published:** 2023-06-13

**Authors:** Tonica Bončina, Srečko Glodež, Brigita Polanec, Lara Hočuršćak, Franc Zupanič

**Affiliations:** Faculty of Mechanical Engineering, University of Maribor, Smetanova 17, 2000 Maribor, Slovenia; tonica.boncina@um.si (T.B.); brigita.polanec@um.si (B.P.); lara.hocurscak@um.si (L.H.); franc.zupanic@um.si (F.Z.)

**Keywords:** POM, PVD coating, metal coating, adhesion analyses, indentation tests

## Abstract

This study presents a comprehensive analysis of different coating materials on the POM substrate. Specifically, it investigated physical vapour deposition (PVD) coatings of aluminium (Al), chromium (Cr), and chromium nitride (CrN) of three various thicknesses. The deposition of Al was accomplished through a three-step process, particularly plasma activation, metallisation of Al by magnetron sputtering, and plasma polymerisation. The deposition of Cr was attained using the magnetron sputtering technique in a single step. For the deposition of CrN, a two-step process was employed. The first step involved the metallisation of Cr using magnetron sputtering, while the second step involved the vapour deposition of CrN, obtained through the reactive metallisation of Cr and nitrogen using magnetron sputtering. The focus of the research was to conduct comprehensive indentation tests to obtain the surface hardness of the analysed multilayer coatings, SEM analyses to examine surface morphology, and thorough adhesion analyses between the POM substrate and the appropriate PVD coating.

## 1. Introduction

In recent decades, engineering components made of different polymeric materials have found extensive use in various engineering applications, offering several advantages over metal components, particularly in office appliances, computer and laboratory equipment, mechatronic devices, household facilities, and medical instruments [1,2,3,4]. Firstly, the mass production of polymeric machine parts and structural components is more cost-effective, especially when utilising injection moulding. Secondly, the use of additional lubrication is not necessary for contacting polymeric machine parts, such as gears and bearings, making them a compelling choice for applications where the use of lubricant is inadequate (e.g., printers, medical instruments) [5,6]. Furthermore, polymers exhibit significant resistance to corrosion, enabling their operation in environments where corrosive substances are found [7,8]. However, polymeric components also possess certain disadvantages compared to most metal components, including lower mechanical properties, lower thermal conductivity, reduced temperature resistance, and decreased manufacturing precision [9,10,11,12].

Polymers are frequently used for manufacturing volumetric elements in the design of dynamically loaded machine components, especially for parts with contacting surfaces such as bearings and gears. Since these components are often used in dry running conditions, i.e., without lubrication, high contact friction is often the main reason for the abbreviated service life due to the extensive wear of the components in the early stage of exploitation [13,14,15,16]. To address this issue, one common approach is to apply low-friction coatings to the polymeric contacting machine elements. This technique, as suggested by Liu et al. [17] and Martinez et al. [18], can not only reduce friction but also improve the surface properties of these components. Dearn et al. [19,20] proposed an approach to reduce friction and wear of polymer gears through the application of various solid lubricant coatings, including molybdenum disulphide (MoS_2_), boron nitride, and polytetrafluoroethylene (PTFE). Their experimental investigation demonstrated that the PTFE coating exhibited the most significant reduction in friction and wear. In another study, Bae et al. [21] investigated the response of contact stress in polymer gears made of PEEK and coated with a diamond-like carbon (DLC) coating. Their numerical results indicated that a DLC coating with a thickness of 2 μm had a relatively small effect on the distribution of contact stress between meshing gears. They explained that the coating thickness was insufficient to significantly influence the bulk deformation characteristics of the polymer gears.

Physical vapour deposition (PVD) is a widely used technique for the deposition of very thin films onto base components to enhance their tribological behaviour (reduced coefficient of friction, lower wear, etc.), optical properties, aesthetic appeal, and other characteristics. Baptista et al. [22] conducted a critical review focusing on process improvement and market trends associated with PVD coatings. The authors concluded that PVD techniques are constantly evolving, keeping pace with the emergence of new technologies adapted to these processes. Additionally, the same authors [23,24] conducted a comprehensive investigation of the wear characterisation of chromium PVD coatings on polymeric substrates for optical components in the automotive industry. Moreover, they highlighted several fields where PVD coatings may be used, including biomedical implants [25], other medical devices [26], the solar industry [27], cutting tools [28,29], and more. In all these applications, PVD processes can involve mono-layered or multi-layered coatings [30,31].

The predominantly used method in PVD is magnetron sputtering, which involves the vaporisation of material through the bombardment of the target with highly energetic ions [32]. Magnetron sputtering utilises a static magnetic field that enables plasma condensation in front of the target, resulting in reduced electric current to the substrate and lower heating. The rate of sputtering is determined by atomic mass, flux density, and ion energy. This method allows for the preparation of hard nanocomposite and multicomponent coatings using various materials. Supplementary technologies, such as plasma activation and plasma polymerisation, are commonly employed in combination with magnetron sputtering to enhance the final properties of the applied PVD coatings [33,34,35]. Due to the low surface energy of polymers, their adhesion to coatings is often inferior. Therefore, it is necessary to activate the polymer surface before the coating process, i.e., magnetron sputtering. Plasma activation is employed to instantly apply a thin layer, preventing the surfaces from losing polarity. In the process of plasma polymerisation, fragments of hydrocarbon, fluorocarbon, and organic molecules are deposited and can accumulate on the substrate surface.

In a previous study [36], the authors examined the wear behaviour of POM spur polymer gears coated with various materials. The experimental observations indicated that the impact of the surface coatings (Al, Cr, and CrN) on the wear behaviour of the POM gears was negligible and could be disregarded. The coatings considered in the study were too thin to affect wear resistance substantially. Furthermore, the analysis of adhesion was not included, which may be a crucial parameter to consider when evaluating the wear behaviour of coated machine parts such as gears. Therefore, a comprehensive adhesion analysis of the Al, Cr, and CrN coatings was necessary to assess the experimental results. Consequently, the aim of this study was to investigate the deposition of multilayer PVD coatings, specifically aluminium (Al), chromium (Cr), and chromium nitride (CrN), on a POM substrate using typical industrial coating parameters. The objective was to assess the feasibility of this approach, which is not commonly explored in the study of metallised polymers. The focus was on the characterisation using scanning electron microscopy and a focused ion beam, as well as the determination of scratch resistance and friction coefficients using the indentation method.

## 2. Materials and Methods

### 2.1. Deposition Process

This research considered three different sets of coatings on a polymer substrate made of polyoxymethylene (POM). The application of aluminium (Al) coating involved a three-step process, including plasma activation, aluminium metallisation through magnetron sputtering, and plasma polymerisation using hexamethyldisiloxane (HMDSO). The chromium (Cr) coating was deposited using a single-step magnetron sputtering process. The chromium nitride (CrN) coating was deposited in two steps: first depositing chromium, followed by reactive deposition of chromium nitride. The coating process parameters (for a single layer) for all the analysed coatings are presented in Table 1. For coatings with several layers, each step of the process outlined in Table 1 was repeated three times (for the three-layer coating) or five times (for the five-layer coating).

The magnetron sputtering process was conducted in a horizontal vacuum chamber with internal dimensions of 570 × 695 × 2970 mm (width × height × length). The weight of the chamber was approximately 100 kg. The charging carriage had a diameter of 496 mm. The minimum target–substrate distance was 100 mm, and the target’s dimensions were approximately 200 × 2300 × 20 mm (width × length × thickness). The volume of the reactor was 1475 L. The chamber was equipped with a glow discharge unit for plasma pre-treatment of the substrates, which was powered by a medium-frequency (MF) power supply rated at 20 kW. Once the plasma pretreatment was finished, the sputtering process began, using a DC-magnetron sputtering source with a maximum power of 200 kW. After the sputtering process concluded, plasma polymerisation began with the deposition of HMDSO as a protective layer. The same MF power supply used for the pretreatment was also used for plasma polymerisation and posttreatment.

During the process, the generator emitted 10,200 to 10,500 kW of energy through the magnetron. According to the set parameters, it reached a power of 200 kW in 19 s and maintained it for a further 43 s. The temperature of the magnetron was regulated during the process, with the magnetron being cooled by water. The temperature of the magnetron ranged from 55 °C (at the start of the sputtering process) to 70 °C (at the end of the sputtering process). If the temperature exceeded the set limits (as shown in Table 1), an error warning was triggered, and the process was stopped; the device then entered standby mode. The temperature for plasma processes was not controlled; however, voltage limits were set between 500 and 5000 V.

The planetarium containing the loaded samples was inserted into the META ROT 500 device, starting the experimental process. In the Al-coating process, the pumping operation commenced in the pre-vacuum mode and was subsequently switched to high-vacuum mode once the desired vacuum level was reached. After 10 s of pumping, the initial pressure of 5 *×* 10^−3^ mbar was reached, and the first process step, namely plasma activation, was initiated. The mass flow controller (MFC) was employed to deliver a controlled flow of air (consisting of 77% nitrogen) into the chamber at a rate of 800 sccm. Following a stabilisation period of 6 s, the plasma generator was switched on and reached a power output of 12 kW within 3 s. The plasma generator was then switched off after operating at this power level for 15 s. PID regulation was utilised to maintain the reference values of airflow and pressure within the set limits by controlling the flow on the MFC. The following step in the process involved the spraying of aluminium. This step was conducted in a high-vacuum environment using argon plasma gas. Within the plasma field, argon atoms were accelerated and collided with the aluminium target. Consequently, the knocked-out aluminium atoms condensed onto the products, which were attached to a planetarium rotating at 50 rpm throughout the process. The resulting coating thickness, achieved at a power of 10,500 kW, ranged from 150 to 200 nm. The magnetron temperature during the process varied between a minimum of 30 °C and a maximum of 90 °C. The magnetron was cooled using warm water, preventing excessive contraction and expansion of the target. As a result, the products were subjected to a temperature rise of approximately 60–80 °C. The final step of the process was plasma polymerisation, which was carried out in a controlled environment with fine (pre)vacuum conditions. Only the (pre)pumps were utilised as not to contaminate the oil in the diffusion pumps. Hexamethyldisiloxane (HMDSO) was introduced into the chamber, where it condensed onto the products. Subsequently, HMDSO underwent polymerisation in the plasma field on the products, forming a thin 20 nm protective layer on the aluminium, effectively preventing oxidation.

The Cr coating process began after 80 s when the initial pressure of 6·10^−4^ mbar was reached. The MFC started to supply 500 sccm of argon to the chamber. After 4 s had passed, the generator was switched on, reaching a power of 120 kW in 40 s and maintaining it for another 65 s before being turned off. During this time, the generator released 10,200 kW of energy through the magnetron. This energy caused the argon atoms to accelerate and collide with the chromium target. The dislodged chromium atoms condensed and adhered to products.

For the CrN coating, the chromium metallisation process began after 90 s, when the initial pressure of 9·10^−4^ mbar was reached. MFC1 began to supply 120 sccm of argon, and MFC2 also started to supply 190 sccm of nitrogen. After 4 s, the generator was switched on. It reached 100 kW of power in 10 s and was switched off after 57 s. During this time, the generator emitted 6200 kW of energy.

### 2.2. Characterisation of Coatings

The uncoated POM substrate surface was inspected using an environmental scanning electron microscope (SEM; Quanta 200 3D, FEI, Eindhoven, the Netherlands). The water pressure inside the chamber was set at 60 Pa. Imaging was conducted using the backscattered electron detector (BSE) and large field detector (LFD) for secondary electrons.

The surfaces of the Al, Cr and CrN-coated samples were thoroughly examined with a field emission scanning electron microscope (SEM; Sirion 400 NC, FEI, Eindhoven, The Netherlands) equipped with an energy-dispersive spectroscopy (EDS) detector for microchemical analysis (INCA x-sight, Oxford Analytical, Bicester, UK). EDS spectra were obtained using a low accelerating voltage of 5 kV to reduce the interaction volume. Cross-section imaging of the coatings was performed using a scanning electron microscope equipped with a focused ion beam (SEM-FIB; Quanta 200 3D, FEI, Eindhoven, the Netherlands). For the milling of the coated POM, a gallium (Ga) ion beam with an accelerating voltage of 30 kV and a beam current of 1 nA was used. Images of the obtained cross-sections were subsequently captured using a Ga-ion beam with low currents ranging from 10 pA to 30 pA at the 30 kV accelerating voltage.

FIB cross-section micrographs were employed to determine the layer thickness using the Quanta 200 3D electron microscope software, considering the tilt correction. The thickness was measured at a minimum of 15 positions in multiple micrographs. Subsequently, the average thickness and standard deviations were calculated.

Additionally, the thickness of the five-layer Al coating was determined using a scanning electron microscope (SEM; JSM-IT800, JEOL, Tokyo, Japan) after preparing the sample in a cross-section polishing machine utilising argon (Ar) ions. The SEM software enables precise measurements of each layer, facilitating accurate determination of the coating thickness. The sample was cut in the cross-section direction using a diamond saw. To protect the Al coating, a 20 min ion sputtering of gold (Au) was performed using the JFC-1100E ion sputtering machine (JEOL, Tokyo, Japan) at 10 mA. Subsequently, the cross-section of the sample was prepared using a cross-section polisher with an accelerating voltage of 2 kV for 6 h and 3 kV for 2 h, while maintaining a chamber temperature of −20 °C. For SEM inspection of the obtained cross-section, the observed surface was coated with carbon using the parameters of a 6 kV accelerating voltage for 5 min.

The surface coverage was estimated using backscattered electron (BSE) micrographs. In the BSE micrographs, the coating appeared bright, while the uncovered POM surface appeared dark. The fraction of the uncoated surface was determined using ImageJ software (ImageJ, National Institutes of Health, Bethesda, MD, USA) by setting an appropriate threshold level.

### 2.3. Indentation Tests and Adhesion Analysis

Nanoindentation testing was performed on both coated and uncoated POM by Nano Test Vantage (Micro Materials Limited, Wrexham, UK). A low load and a high load system of the instrument have maximum load ranges of 0.1 mN to 500 mN and 500 mN to 30 N, respectively. The indentation hardness and indentation modulus of the selected material were determined by indenting the three-sided pyramid (Berkovich) diamond indenter into the chosen material. The indentation process was controlled by an electromagnetic drive loading system equipped with a high-precision coil and a permanent magnet. The indentation testing involved applying increasing loads to examine the indentation properties. In the low-load system, a test series was conducted by increasing the load from 1 mN to 5 mN at a fixed location. Each test consisted of a 20-s loading period from 1 mN to 5 mN, followed by a 10 s hold at the maximum load of 5 mN, and finally a 10 s unloading period. The indentation curves were corrected for thermal drift, and the diamond area function of the Berkovich indenter (area *A* = 500 *h* + 23.5 *h*^2^, where *h* is the depth of the indentation) was applied for calculating the indentation properties.

Scratch testing was conducted to examine the adhesion properties of the various coatings on POM using a Nano Test Vantage with a spherical diamond tip of 23 μm diameter, applying loads from 0.1 mN up to 250 mN. During each test, five scratches were made at a scanning velocity of 10 μm/s. The scratch process began with a low load of 0.1 mN for the initial 50 μm length, followed by a gradual increase in load at a rate of 10 mN/s until reaching the maximum load of 250 mN for the subsequent 250 μm length. Finally, a constant maximum load of 250 mN was applied for the remaining 50 μm length of the scratch. The topography of the surface was measured before and after the scratch test to determine the difference between the initial roughness of the (un)coated POM and the topography measured during and after the scratch test. The critical load required to cause the damage was subsequently determined by inspecting the scratched surfaces using SEM, following the guidelines outlined in ASTM C1624 [37] and EN ISO 20502:2016 [38] standards.

The friction between the tip and the sample was measured using a friction probe to determine the coefficient of friction. The coefficient of friction is calculated as the ratio of the measured friction force to the normal force (load). The measurement was performed using the NanoTest Vantage system with a spherical diamond tip of 23 μm diameter. During each test, five scratches were made at a scanning velocity of 1 μm/s. The initial 50 μm length of the scratch was subjected to a very low load of 0.1 mN, followed by an abrupt increase to the maximum load of 5 mN, which was then applied for the next 500 μm. The topography of the surface was measured before and after the friction probe test with a load of 0.1 mN at the exact testing location. This allowed for the subtraction of the initial roughness from the topography measured during and after the friction probe test.

## 3. Results and Discussion

### Coating Characterisation

Figure 1 displays the photography of all the deposited samples (30 mm × 10 mm × 5 mm) used in the investigation. Macroscopically, the coatings appear uniform, and the original white colour of the POM has transformed into various shades of grey. Traces of machining can be seen on the surface.

Figure 2, Figure 3 and Figure 4 depict the surface appearance (backscattered electron images) and FIB cross-sections (IISE—ion-induced secondary electrons) of the coatings. All three Al-coated samples exhibited uniform coatings (Figure 2a,c,e). In contrast, the Cr coating was non-uniform and appeared broken, resembling fish scales (Figure 3a,c,e). The CrN coating covered the surface of the POM well with a single layer, but the coverage fraction decreased with additional CrN layers (Figure 5a,c,e).

The surface appearances of the one-layer and three-layer Al coatings were almost identical (Figure 2a,c). However, the surface with the five-layer Al coating was rougher, showing numerous growth defects. The FIB cross-sections were used to measure the thicknesses of the layers, which were approximately 300 nm, 650 nm and 1200 nm for one, three and five layers of Al, respectively.

It is important to note that the thickness of multilayers is not a simple sum of the number of layers due to the repeated deposition cycle and surface activation, which remove the previous layer’s surface. The single-layer coating exhibited good adhesion to the POM substrate, while the three-layer coating appeared compact with visible interfaces between the layers. As for the five-layer coating, porosity was observed between the individual layers, possibly caused by layer decohesion due to thermal stresses induced by repeated cooling and heating.

The porosity can also be an artefact of preparation. The cross-section prepared in a cross-section polishing machine utilising argon (Ar) ions is shown in Figure 3. This shows uniform layers of aluminium (thickness 200–250 nm) with HMDSO interlayers with thicknesses in the range of 50 nm.

The results of the microchemical analysis (EDS) are presented in Table 2. It can be observed that only in the case of one coating did the interaction volume reach the polymer substrate, resulting in significantly higher contents of carbon (C) and oxygen (O). The outermost layer of HMDSO contributed to the signals of C, O and Si. The detection of oxygen suggests the possible presence of a small amount of aluminium oxide in the layer.

The Cr layers, as observed in Figure 4, exhibited a scale-like structure with sizes typically ranging from 20 to 30 μm. These scales did not adhere well to the surface, which is evident in Figure 5 as well. The Cr layer experienced breakage and subsequent peeling off from the surface. Additional deposition of Cr layers did not improve the adhesion. The surface coverage of the Cr coating was only around 33%. The thickness of a single Cr layer was approximately 200 ± 140 nm, while the thickness of the coating varied significantly for three or five layers (with an average thickness of around 410 nm for both cases). Table 3 provides further details on the coating thickness, with notable deviations exceeding 100 nm. The EDS analysis revealed that the Cr layers primarily consisted of Cr, with some C and O originating from the substrate (refer to Table 2). Additionally, the spherical growth on the surface of the scale exhibited traces of iron (Fe).

A single layer of CrN appeared uniform on the BSE image (Figure 6a). However, the micrograph taken by secondary electrons induced by the ion beam revealed that the surface was not fully covered (Figure 7). The peeling off mainly occurred at ridges created during machining. The percentage of uncovered surface area increased from approximately 4% in the single-layer coating to 10% in the three-layer coating. In the case of the five-layer coating, only 50% of the surface was covered. These findings suggest that the multilayer deposition of CrN is not advantageous, and that the POM surface should be flat and smooth. The thickness of the layers was measured to be approximately 170–250 nm, 300–400 nm and 400–540 nm for one, three and five layers, respectively. The EDS analysis revealed the presence of Cr, N, O and C (see Table 2). The presence of C and O is likely attributed to the substrate.

The indentation hardness and modulus of POM and different types of coatings are presented in Table 3. The POM substrate is softer than any coating material used. A load between 1 and 5 mN also caused plastic deformation of the substrate, which decreases with increasing coating thickness. Therefore, it is expected that increasing the number of coatings will result in increased hardness if the coating adheres well to the surface. This is evident in the case of Al coatings, where both hardness and modulus increase, approaching the hardness of pure Al in a five-layer coating. Nevertheless, the values of indentation hardness and reduced modulus should be taken as a combined response of both coating and substrate. The measurement of the exact coating hardness would require much smaller loads, which can also bring out the indentation size effect at smaller indentation depths and should be combined with numerical simulations.

A slight increase in both properties was also observed in the case of CrN. However, the values are relatively low, generally lower than the hardness and modulus of Al coatings and significantly lower than the properties of pure CrN (11.2 GPa). The one-layer Cr coating exhibits much higher hardness than any other coating (the hardness of bulk Cr is 1.225 GPa). It should be noted that the hardness was measured in areas where Cr adhered rather well to the substrate, while in uncovered regions, the properties were even lower than the original POM, possibly due to the degradation of POM properties during thermal exposure.

Table 4 presents the average values of the coefficient of friction for POM and coated samples. The coatings with Cr and CrN did not significantly reduce the coefficient of friction, while the coating with five layers of Al reduced the coefficient of friction by more than 50%.

Figure 8 presents typical experimental results. The low normal load of 5 mN primarily induced elastic deformation in the specimen. The waviness resulting from the surface topography is evident. Only the deposition with five Al layers significantly reduces the friction coefficient. Therefore, this coating configuration is expected to be advantageous in reducing the sliding wear rate. There are several factors that warrant further investigation. Firstly, the Al coating exhibits a higher level of uniformity compared to other coatings. Secondly, the five-layer Al coating has a thickness exceeding 1 μm, which helps in reducing surface macro-roughness. Thirdly, due to its uniformity and thickness, it induced less plastic deformation in the POM substrate. Fourthly, the presence of an HMDSO protective layer in the Al coating may contribute to friction reduction. Lastly, the micro-level waviness of the coating may cause the moving diamond sphere to come into contact with surface asperities, potentially reducing friction.

Figure 9 depicts the five scratches on each of the coated surfaces and compares them with the uncoated POM surface. The Al-coated samples predominantly exhibited crack formation as the primary type of damage (Figure 9a,b). The distance from the starting point at which the initial cracks appeared decreased proportionally with the increasing number of coating layers. The distance to the point of breakthrough was measured for each crack based on the images. Generally, the cracking of the coating is initiated either at the edge or in the centre of the indenter, following the direction of the scratch. The coating then began to peel off from the POM, either in the direction of the scratch or laterally, indicating inadequate adhesion between the various coating types and the substrate. The non-uniformity of the Cr coatings (Figure 9c,d and Figure 10) resulted in them not significantly contributing to protection against scratching. A single layer of CrN performed slightly better than a single layer of Al, but the superiority of Al layers over multiple CrN layers was evident (Figure 9e,f).

Figure 11 illustrates the maximum depths under load and the maximum plastic depths observed after the indenter penetrated through the coatings. It is evident that the coatings did not enhance the resistance to penetration.

Figure 12 presents the depth profiles of different scratches for various types of coating materials. For the given number of layers, the occurrence of surface damage appeared at a similar distance from the starting point. The profile of the scratch on the uncoated substrate material is also included. Plastic deformation in the uncoated POM started at approximately 20 mN and increased until reaching a distance of 200 nm, where a pop-in phenomenon occurred due to substrate smearing. The coatings have a minimal effect on the load behaviour (Figure 12a,c,e), except for the three-layer Al coating. The presence of three and five Al layers reduced plastic deformation at lower loads, while Cr and CrN coatings even deteriorated these properties.

The results indicate that applying multilayer coatings to a POM substrate using typical industrial coating parameters may not necessarily yield beneficial effects. This approach has shown viability for soft and ductile metals such as aluminium. One of the primary challenges lies in the differential thermal expansion between the substrate and coating, which can generate significant internal stresses in the coating. Ductile aluminium coatings can accommodate these stresses through plastic deformation. On the other hand, Cr, being a hard and brittle transition metal, and CrN, a brittle ceramic material, are more susceptible to fracture during thermal cycling. However, our previous work and current findings highlight the need to enhance the adhesion between Al and POM to improve resistance to peeling. In our opinion, adhesion could be enhanced by optimising the plasma surface activation step.

The one-layer CrN coating exhibits good adhesion to the POM substrate. It would be advisable to test the one-layer CrN coating with varying thicknesses to determine the optimal value in terms of scratch resistance and coefficient of friction. However, the application of multiple layers is not effective. Decohesion at the coating–substrate interface occurs due to repeated heating and cooling, and as the number of cycles increases, more and more coating is removed from the surface, resulting in a decrease in the coverage fraction.

The metallisation of POM with Cr has proven to be the most critical. Even with a one-layer Cr coating, the application of typical industrial parameters has shown inadequacy. However, since chromium has been successfully applied to various polymer substrates, it is possible to achieve successful metallisation of POM as well. It is likely that the current thickness of Cr is above the optimal value. There are several parameters in magnetron sputtering that can be adjusted accordingly. The most important ones are sputtering power, substrate temperature, minimum and maximum temperature during sputtering, sputtering rate, substrate rotation speed and plasma surface activation. Furthermore, it is believed that the flatness and roughness of the POM substrate significantly impact the metallisation process. Therefore, additional attention should be given to post-machining processes to ensure a smoother surface. Future investigations should prioritise improving the adhesion of different coatings to the POM substrate, optimising the thickness of a single layer, and preventing coating fracture during repeated metallisation processes.

As mentioned earlier, the deposition of an Al-multilayer coating using a typical industrial process results in a nanolayer structure consisting of sequential layers of metallic aluminium and polymeric HMDSO. The results of this study demonstrate that the deposition of HMDSO on Al yields good adhesion at the interface, which is consistent with previous findings. Furthermore, the same phenomenon occurred in the inverse situation, when depositing Al on HMDSO. Therefore, by repeating the deposition of aluminium layers, a new type of nanolayer coating with beneficial effects on several properties can be produced. With an increasing number of cycles, the hardness increases, and to a lesser extent, the modulus reduces. It may be even better to reduce the thickness of Al layers below 100 nm, as it may enhance both the hardness and toughness of the deposit. Additionally, increasing the thickness of the coating enhances its mechanical strength, thereby improving its resistance to scratching. The reason for the decrease in the friction coefficient is not entirely clear, considering that the top layer is always HMDSO. However, it could be attributed to a decrease in surface roughness.

## 4. Conclusions

The present study aimed to examine the deposition of multilayer PVD coatings of aluminium (Al), chromium (Cr) and chromium nitride (CrN) on a POM substrate. The primary focus was characterising the coatings using scanning electron microscopy and focused ion beam, as well as assessing their scratch resistance and friction coefficient using the indentation procedures. Based on the experimental findings, the following conclusions can be drawn:The coatings exhibited macroscopic uniformity. On the microscale, only the Al coatings displayed uniformity and complete coverage of the surface. The one-layer CrN coating also exhibited uniformity, but the coverage decreased with an increase in the number of CrN layers. The Cr coating, however, displayed significant non-uniformity.The coefficient of friction did not show any significant changes with the application of coatings, except in the case of Al coatings. A five-layer Al coating resulted in a reduction of over 50% in the coefficient of friction compared to uncoated POM.The scratch resistance was significantly improved in Al-coated POM, while the other coatings did not provide substantial improvement.The typical industrial process used for depositing multiple layers was found to be unsuitable for the deposition of Cr and CrN coatings. Optimal parameters should be determined for each specific coating material.

## Figures and Tables

**Figure 1 materials-16-04365-f001:**
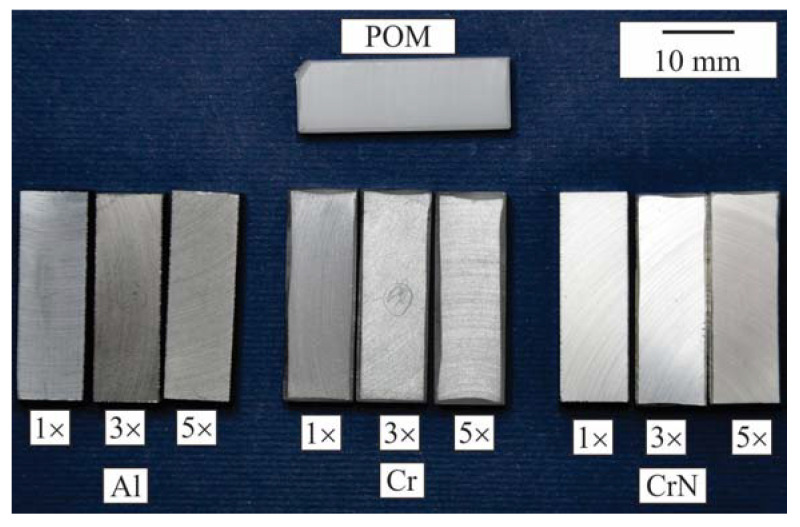
Photograph of the investigated samples.

**Figure 2 materials-16-04365-f002:**
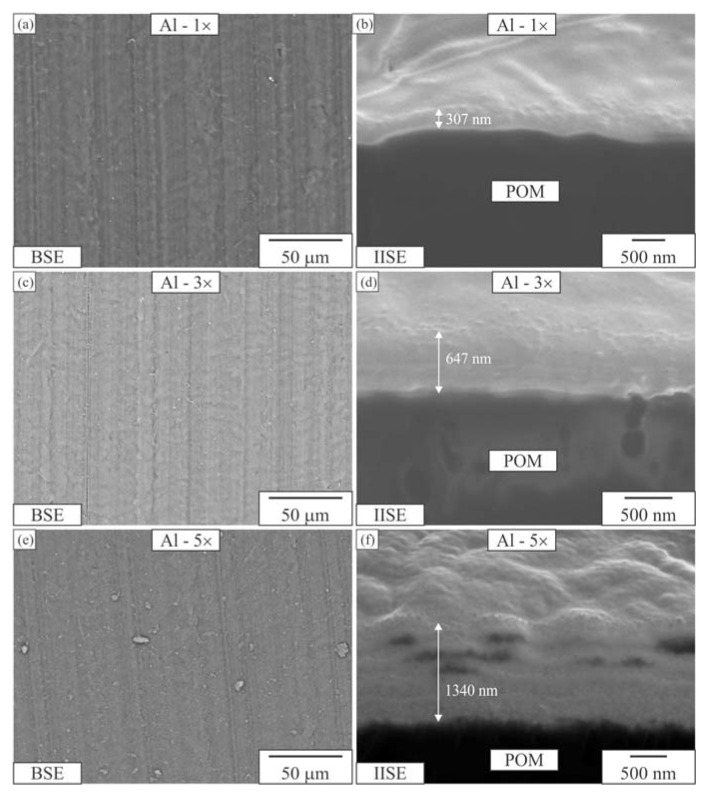
Electron micrographs of Al coatings on the POM substrate. One Al layer: (**a**) coating surface, (**b**) FIB cross-section. Three Al layers: (**c**) coating surface, (**d**) FIB cross-section. Five Al layers: (**e**) coating surface, (**f**) FIB cross-section. BSE—backscattered electron micrograph, IISE—ion-induced secondary electron micrograph.

**Figure 3 materials-16-04365-f003:**
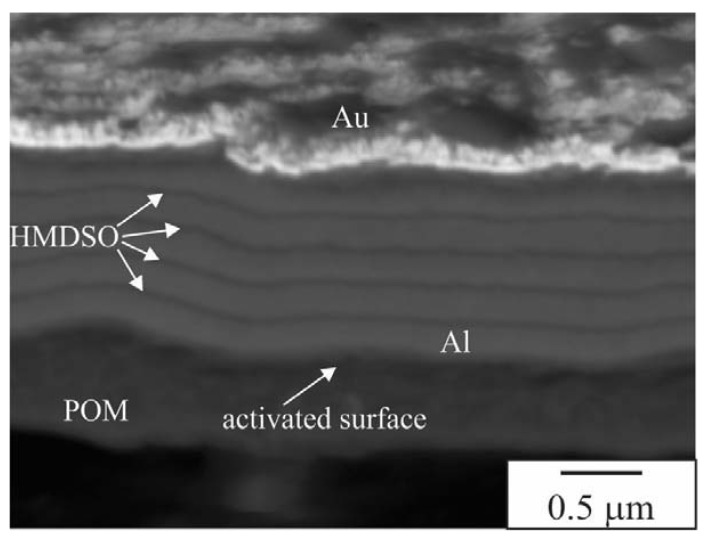
A high-resolution electron micrograph (UHD—upper hybrid detector) of Al coating with five layers. The cross-section was prepared in a cross-section polishing machine utilising argon (Ar) ions.

**Figure 4 materials-16-04365-f004:**
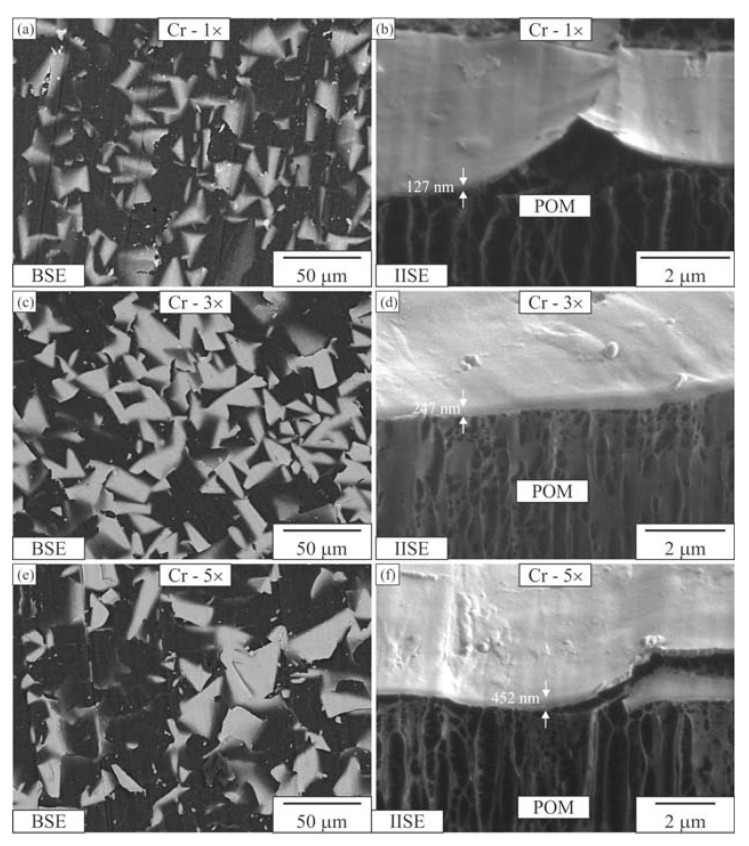
Electron micrographs of Cr coatings on the POM substrate. One Cr layer: (**a**) coating surface, (**b**) FIB cross-section. Three Cr layers: (**c**) coating surface, (**d**) FIB cross-section. Five Cr layers: (**e**) coating surface, (**f**) FIB cross-section. BSE—backscattered electron micrograph, IISE—ion-induced secondary electron micrograph.

**Figure 5 materials-16-04365-f005:**
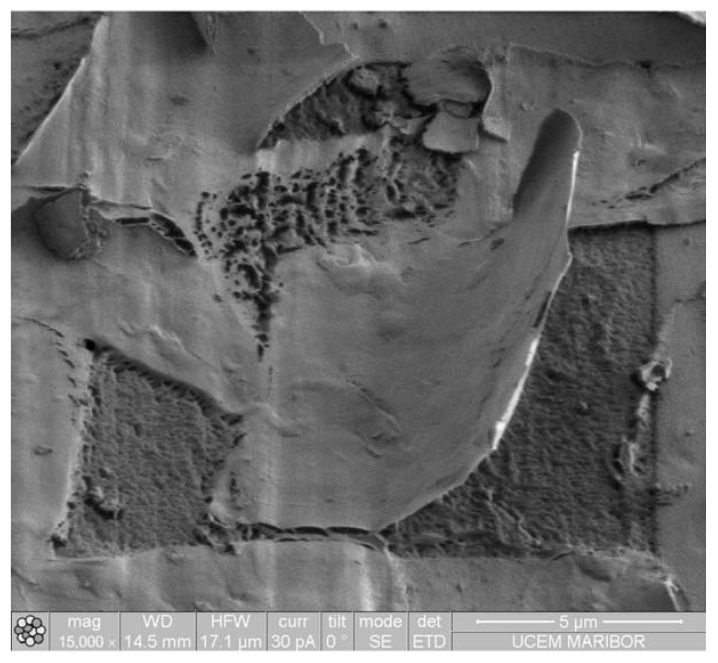
IISE micrograph of the one-layer Cr coating on the POM substrate.

**Figure 6 materials-16-04365-f006:**
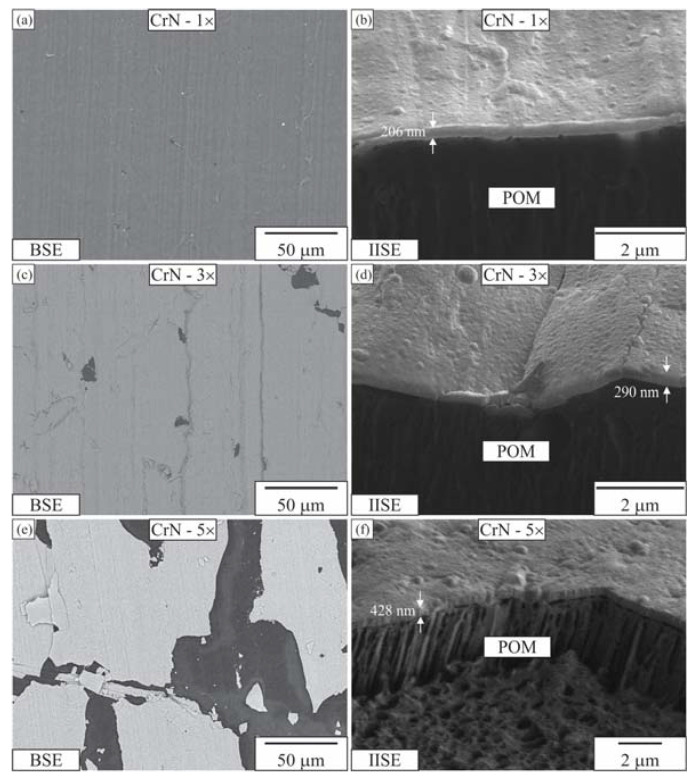
Electron micrographs of CrN coatings on the POM substrate. One CrN layer: (**a**) coating surface, (**b**) FIB cross-section. Three CrN layers: (**c**) coating surface, (**d**) FIB cross-section. Five CrN layers: (**e**) coating surface, (**f**) FIB cross-section. BSE—backscattered electron micrograph, IISE—ion-induced secondary electron micrograph.

**Figure 7 materials-16-04365-f007:**
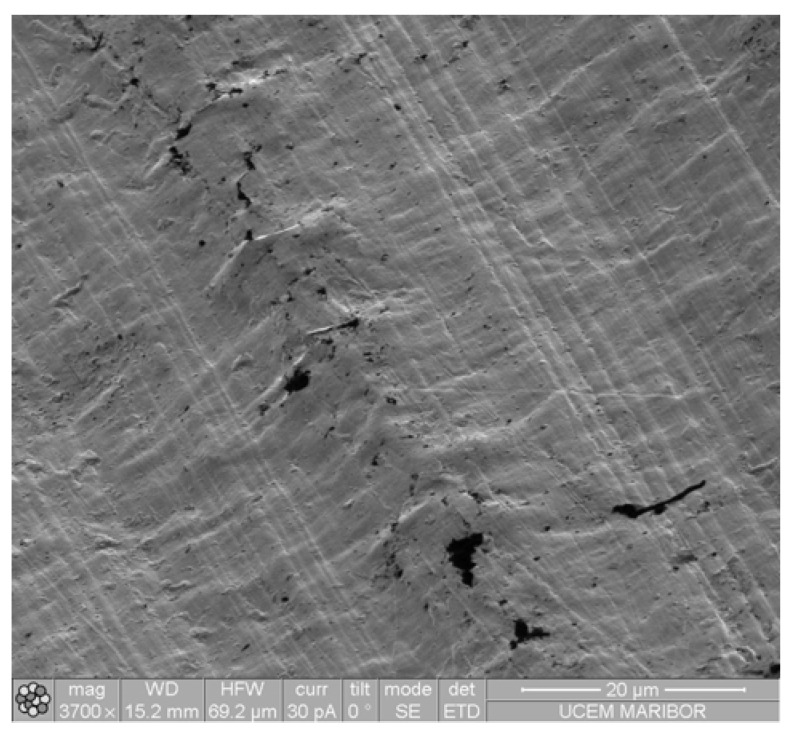
IISE micrograph of the one-layer CrN coating on the POM substrate.

**Figure 8 materials-16-04365-f008:**
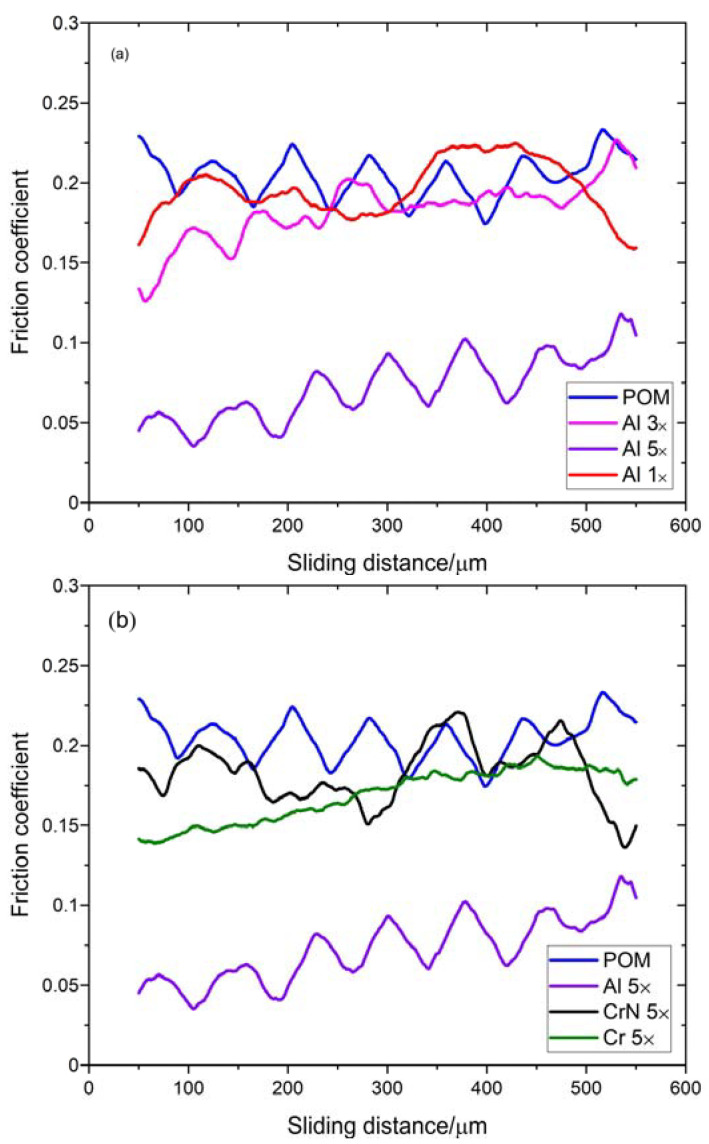
Coefficient of friction: (**a**) comparison of POM and Al coatings, (**b**) comparison of POM and five-layer coatings of Al, Cr and CrN. Diamond sphere, 23 μm diameter, load 5 mN.

**Figure 9 materials-16-04365-f009:**
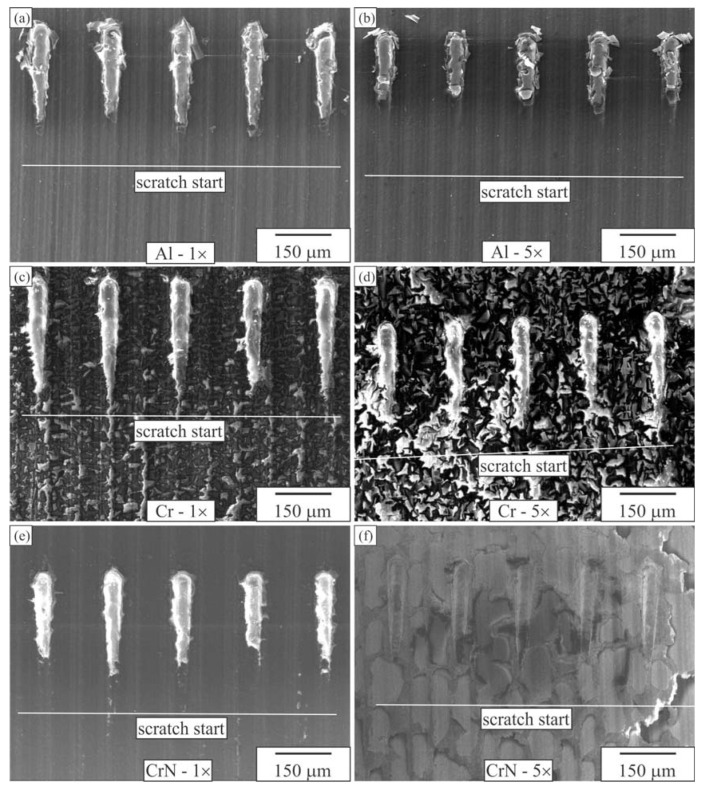
Scratched surfaces of POM coated with 1 and 5 layers of Al, Cr and CrN (BSE micrographs). One-layer (**a**) Al coating, (**c**) Cr coating, (**e**) CrN coating, five-layer (**b**) Al coating, (**d**) Cr coating, (**f**) CrN coating.

**Figure 10 materials-16-04365-f010:**
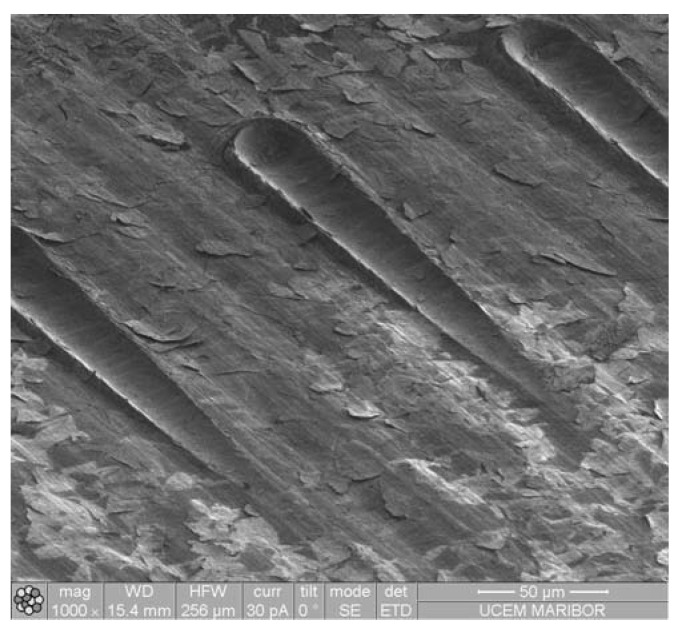
IISE micrograph of scratches on Cr coating (5 layers).

**Figure 11 materials-16-04365-f011:**
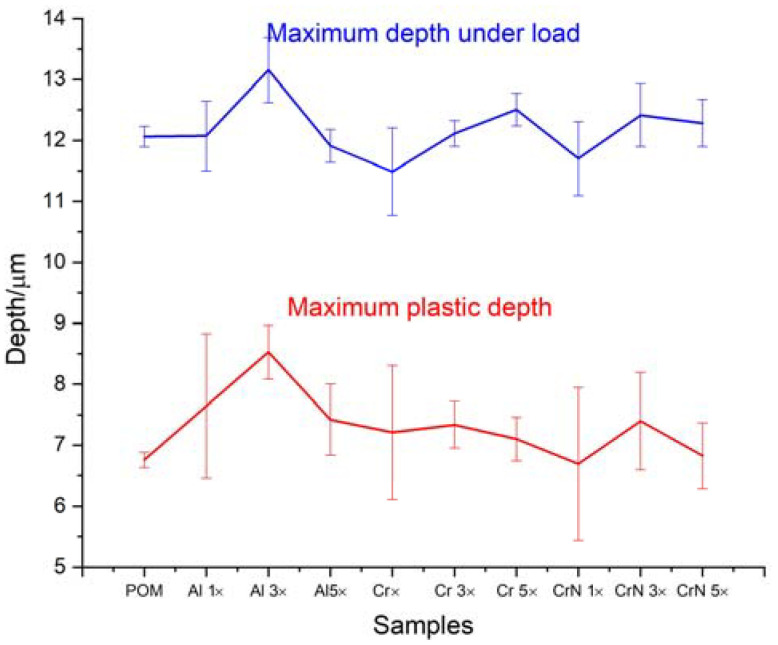
Diagram illustrating the maximum depth reached during testing under the applied load (upper, blue line), and the topography observed after testing, indicating the plastic depth (lower, red line).

**Figure 12 materials-16-04365-f012:**
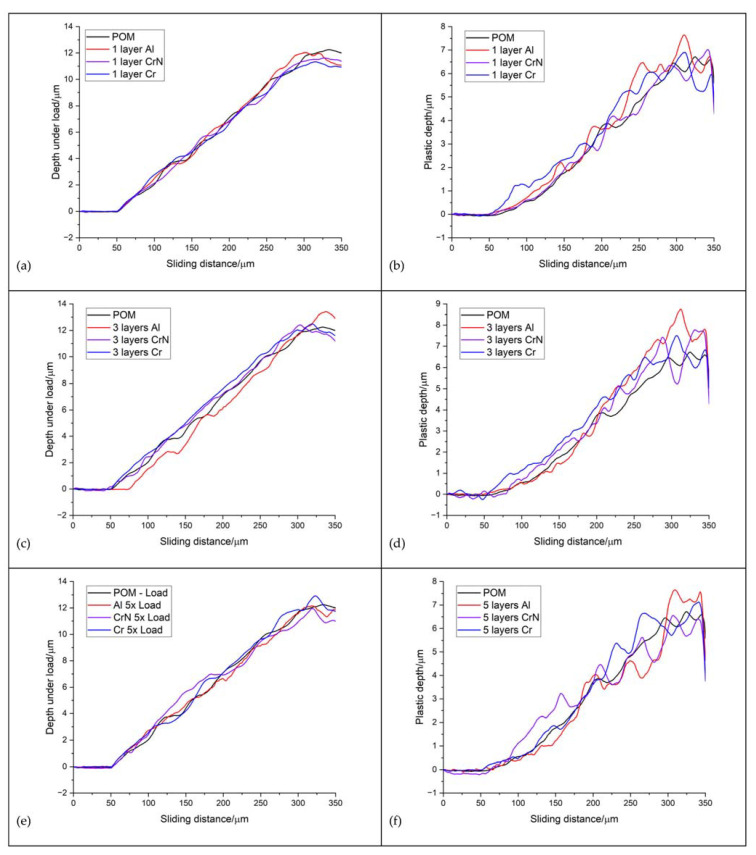
Diagram depth versus sliding distance for (**a**,**b**) one-layer coatings, (**c**,**d**) three-layer coatings and (**e**,**f**) five-layer coatings.

**Table 1 materials-16-04365-t001:** The process parameters for the analysed coatings (one layer).

Coating	Process	Pumping Time [s]	Starting Pressure [mbar]	Mass Flow Contr.	Regulation Pressure [mbar]	Process Time [s]	Regulation Energy [kWs]	*T* [°C]	
								Min	Max
**Al**	Plasma activation	10	5·10^−3^	800	3·10^−2^	18	198	-	-
	Magnetron sputtering	150	4·10^−4^	500	2.2·10^−3^	62	10,500	30	90
	Plasma polymerisation	1	1.5·10^−2^	300	2·10^−2^	50	582	-	-
**Cr**	Magnetron sputtering	80	6·10^−4^	500	3·10^−3^	105	10,200	25	90
**CrN**	Magnetron sputtering	80	6·10^−4^	500	3·10^−3^	105	10,200	25	90
	Reactive metallisation	90	9·10^−4^	120	3.4·10^−3^	67	6200	40	90

**Table 2 materials-16-04365-t002:** The chemical composition of the deposited layer as determined using EDS at a 5 kV accelerating voltage (in at. %).

Surface Layer	C	N	O	Al	Si	Cr	Fe
POM-1 layer Al	37.1 ± 5.2		54.7 ± 2.4	7.9 ± 1.3	0.3 ± 0.2		
POM-3 layers Al	18.3 ± 6.3		42.4 ± 1.3	36.8 ± 2.0	2.5 ± 1.5		
POM-5 layers Al	20.4 ± 2.1		25.8 ± 1.9	50.4 ± 2.2	3.4 ± 1.7		
POM-1 layer Cr	47.4 ± 6.2		26.7 ± 2.4			26.0 ± 1.4	1.3 ± 0.6
POM-3 layers Cr	22.5 ± 4.1		16.8 ± 1.4			60.7 ± 6.0	2.5 ± 1.2
POM-5 layers Cr	7.7 ± 2.7		16.5 ± 2.1			76.2 ± 3.6	2.4 ± 0.7
POM-1 layer CrN	45.6 ± 6.0	10.3 ± 2.2	38.2 ± 5.8			5.9 ± 11.4	
POM-3 layers CrN	22.8 ± 5.9	24.0 ± 3.6	35.5 ± 0.9			17.8 ± 2.2	
POM-5 layers CrN	9.6 ± 1.0	22.5 ± 3.1	30.5 ± 2.5			37.5 ± 3.6	

**Table 3 materials-16-04365-t003:** Coating thickness, indentation hardness and modulus in the range between 1 mN and 5 mN.

Coating	Coating Thickness [nm]	Hardness [MPa]	Reduced Modulus [GPa]
POM	-	100 ± 1	3.2 ± 0.2
POM-1 layer Al	301 ± 16	172 ± 33	5.2 ± 0.6
POM-3 layers Al	657 ± 26	214 ± 26	5.2 ± 1.1
POM-5 layers Al	1188 ± 225	318 ± 63	6.0 ± 1.2
POM-1 layer Cr	201 ± 141	765 ± 116	8.3 ± 0.9
POM-3 layers Cr	417 ± 124	135 ± 13	3.7 ± 0.3
POM-5 layers Cr	413 ± 88	91 ± 9	2.9 ± 0.1
POM-1 layer CrN	209 ± 21	121 ± 11	2.5 ± 0.1
POM-3 layers CrN	342 ± 34	131 ± 16	4.4 ± 0.7
POM-5 layers CrN	454 ± 45	167 ± 30	5.1 ± 0.7

**Table 4 materials-16-04365-t004:** Measured values of the coefficient of friction.

Surface Layer	One Layer	Three Layers	Five Layers
POM		0.22 ± 0.05	
Al	0.21 ± 0.05	0.16 ± 0.12	0.09 ± 0.05
Cr	0.24 ± 0.08	0.25 ± 0.09	0.16 ± 0.04
CrN	0.24 ± 0.18	0.23 ± 0.06	0.20 ± 0.08

## Data Availability

The data presented in this study are available on request from the corresponding author.

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
