# Peer review of "Comprehensive Analysis of Different Coating Materials on the POM Substrate"

_materials, 2023, doi:10.3390/ma16124365_

Round 1

Reviewer 1 Report

(1)   Section 3.1, Paragraph 3, Line 3: It is difficult to follow how the thickness is measured and the calculation of error. Is it possible to mark the thickness in the cross-sectional images?

(2)   Section 3.1, Paragraph 3, Line 11: Better to use atomic percentage for the coating compositions. In addition, elemental composition for 1x, 2x, and 5x should be shown to compare the composition changes or prove the statement “not many differences”.

(3)   Section 3.1, Paragraph 4, Line 3: I suggest adding the content for how the surface coverage was measured. The same question arises at the surface coverage descriptions.

(4)   Section 3.1, Paragraph 4, Line 3 “The EDS showed that Cr layers contained mainly Cr, with some C and O from the substrate. In the spherical growth on the surface of the scale some Fe was also detected.” The EDS results are not shown.

(5)   For Table 2, please explain why the hardness for Al and CrN coating increases with the increasing number of layers, while Cr coating decreases.

Author Response

Comment #1: Section 3.1, Paragraph 3, Line 3: It is difficult to follow how the thickness is measured and the calculation of error. Is it possible to mark the thickness in the cross-sectional images?

Response: The following paragraph regarding the thickness measurements has been added in Section 2.2 of the revised manuscript:

FIB cross-section micrographs were used for the determination of layer thickness using the Quanta 200 3D electron microscope software, taking into account the tilt correction. The thickness was measured at a minimum of 15 positions in several micrographs. Thereafter, the average thickness and standard deviations were calculated.

Comment #2: Section 3.1, Paragraph 3, Line 11: Better to use atomic percentage for the coating compositions. In addition, elemental composition for 1x, 2x, and 5x should be shown to compare the composition changes or prove the statement “not many differences”

Response: Thank you for this remark. We added a new table (Table 2), which shows the composition of the coatings determined using EDS. The results are given in %.

Comment #3: Section 3.1, Paragraph 4, Line 3: I suggest adding the content for how the surface coverage was measured. The same question arises at the surface coverage descriptions

Response: The following paragraph has been added in Section 2.2 of the revised manuscript:

The surface coverage was estimated using backscattered electron (BSE) micrographs. In the BSE micrographs, the coating appeared bright, and the uncovered POM was dark. The fraction of the uncoated surface was determined using ImageJ by setting an appropriate threshold level.

Comment #4: Section 3.1, Paragraph 4, Line 3 “The EDS showed that Cr layers contained mainly Cr, with some C and O from the substrate. In the spherical growth on the surface of the scale some Fe was also detected.” The EDS results are not shown.

Response: Thank you for the remark. A new Table 2 collects all results of EDS analyses.

Comment #5: For Table 2, please explain why the hardness for Al and CrN coating increases with the increasing number of layers, while Cr coating decreases”.

Response: Thank you for this comment. We revised the text before Figure 5 as follows.

In table 3, the indentation hardness and modulus for the uncoated POM and for the three different types of coatings with different thicknesses are shown. The POM substrate is softer than any type of coating material. A load between 1 and 5 mN caused also the plastic deformation of the substrate, which is reduced with increasing coating thickness. Thus, increasing hardness with the increasing number of coatings is expected if the coating adheres well to the surface. This clearly happened with Al-coatings, where both hardness and modulus increased, approaching the hardness of pure Al in a five-layer coating. A slight increase in both properties was also observed in the case of CrN. However, the values are rather low, generally lower than the hardness and modulus of Al-coatings and much lower than the properties of pure CrN (11.2 GPa ). One-layer Cr coating has much higher hardness than any other coating (the hardness of bulk Cr is 1225 MPa). It has to be stressed that the hardness was measured at places where Cr adhered to the substrate rather well, while at uncovered regions, the properties were even lower than in the original POM, probably due to degradation of POM properties during thermal exposure.

Reviewer 2 Report

This is very interesting paper about the features of different coating materials on the POM substrate. The results are well presented and conclusions are well supported by the experimental results. The reviewer would like to recommend this paper to be published if the following issues can be clarified.

1. As for the preparation of coatings of Al, Cr, and CrN by PVD method, why their processes are different ?( three step processing for Al coating, one step for Cr coating, two steps for CrN coating)

2. Why the authors select Al, Cr, and CrN coatings as researching object, rather than others?

3. The scale bar should be added into the Fig.1.

4. In Fig.7, why the COF of Al 5× coating is the lowest in both Fig.7(a) and (b)?

5. The wear rate or wear volume results should be added into the manuscript to characterize the wear performance of the coating.

6. Why the degree of covering decreased with the number of CrN layers?

7. Why other coatings did not improve the scratch resistance except for Al coating?

Author Response

This is very interesting paper about the features of different coating materials on the POM substrate. The results are well presented and conclusions are well supported by the experimental results. The reviewer would like to recommend this paper to be published if the following issues can be clarified:

Comment #1: As for the preparation of coatings of Al, Cr, and CrN by PVD method, why their processes are different ? (three step processing for Al coating, one step for Cr coating, two steps for CrN coating).

Response: Distinct processes are required, preferentially due to different adherence of these coating types on the polymer substrate. Aluminium typically adheres poorly to the substrate. Therefore, the activation of the polymer surface is required. In our case, we applied nitrogen, which forms covalent bonds with aluminium. The last step is the protection of Al with hexamethyl disiloxane, which prevents the formation of an oxide layer on the aluminium coating. The oxide layer may deteriorate the optical properties of Al. Chromium typically adheres well to polymer surfaces. So, it can be applied in one step. CrN cannot be applied directly to the polymer surface because it does not adhere to it. Therefore, a thin Cr interlayer is required. 

Comment #2: Why the authors select Al, Cr, and CrN coatings as researching object, rather than others?

Response: As already presented in Ref. [22-24], Physical Vapour Deposition (PVD) coatings by Al, Cr or CrN on the polymeric substrate is a well-known technology that is widely used for the deposition of thin films regarding many demands, namely optical enhancement, visual/esthetic upgrading, and many other fields, with a wide range of applications already being perfectly established. For that reason, the Al, Cr and CrN PVD coatings were chosen to obtain the surface hardness of analysed multilayer coatings, surface morphology and adhesion behaviour between POM substrate and appropriate PVD coating. Namely, these characteristics play a crucial role when analysing the wear behaviour of rolling/sliding polymeric mechanical elements (i.e. gears) coated by different PVD coatings.    

Comment #3: The scale bar should be added into the Fig.1

Response: Thank you for the remark. We added the scale bar to Fig. 1.

Comment #4: In Fig.7, why the COF of Al 5× coating is the lowest in both Fig.7(a) and (b)?

Response: Thank you for the remark. We added the following text before Table 4 in the revised manuscript:

There are several reasons which should be investigated by further research. First, Al-coating is rather uniform in comparison to other coatings. Second, the Al with five layers is thicker, more than 1 mm; thus, it reduces the surface macroroughness. Third, since it is uniform and thick, it produces less plastic deformation of the substrate POM. Four, it has a HMDSO protective layer, which may reduce friction. Five, it also has waveness on a microlevel, causing the moving diamond sphere touches the surface at asperities and possibly reduce the friction.

Comment #5: The wear rate or wear volume results should be added into the manuscript to characterise the wear performance of the coating?

Response: As explained in the Introduction, the proposed manuscript is the continuation of the Authors’ previous work [36], where the wear behaviour of coated spur polymer gears made of POM was investigated. In work [36], we found out that the influence of the considered surface coatings (Al, Cr, and CrN) on the wear behaviour of the POM gears is very small and can be neglected. This finding was attributed to the fact that the considered coatings were very thin, thus not impacting the wear resistance substantially. Furthermore, the adhesion analysis was not proposed, which may be a crucial influencing parameter when analysing the wear behaviour of coated machine parts like gears. For that reason, a comprehensive adhesion analysis of considered Al, Cr, and CrN coatings was needed to evaluate the obtained experimental results, which is actually the main goal of the proposed manuscript.

Comment #6: Why the degree of covering decreased with the number of CrN layers.

Response: Thank you for the question. We included the discussion regarding this issue in the section Results and Discussion as follows:

One-layer CrN coating adheres rather well to the POM substrate. It would be sensible to test the one-layer CrN coating with different thicknesses to find the optimal value regarding, e.g. scratch resistance and coefficient of friction. The application of several layers does not work well. The decohesion at the coating–substrate interface occurred by repeating heating and cooling, and with the increasing number of cycles, more and more coating is removed from the surface, thus decreasing the covering fraction.

Comment #7: Why other coatings did not improve the scratch resistance except for Al coating.

Response: No other coatings were continuous; there were coated and uncoated areas. Thus in uncoated areas, the properties were even worse than in the initial POM because of their degradation of properties during thermal cycling.

Reviewer 3 Report

The authors have given a good piece of work with the formation of multilayers on POM substrate, using PVD technique.

In Pg 2, Line No.46 MoS2 has to written as MoS2

Fig 4, shows The coating is not uniform, in the case of multi layer deposition. The adhesive nature is been monitored by some other factors. How to overcome this issue

The authors can add few sentences regarding te application part of these type of coatings which is more useful to the scientific community.

Author Response

The authors have given a good piece of work with the formation of multilayers on POM substrate, using PVD technique.

Comment #1: In Pg 2, Line No.46 MoS2 has to written as MoS2?

Response: It was corrected.

Comment #2: Fig 4 shows the coating is not uniform, in the case of multi layer deposition. The adhesive nature is been monitored by some other factors. How to overcome this issue.

Response: Thank you for the suggestion. We included the discussion regarding this issue in the section Results and Discussion as follows:

The results showed that the application of multilayer coatings on POM substrate using typical industrial coating parameters does not necessarily bring beneficial effects. It was shown that this approach could be viable for soft and ductile metals, such as aluminium. One of the basic problems can be dilatation during repeated heating and cooling. Different thermal expansion coefficients of the substrate and coating can lead to the formation of large internal stresses in the coating. The stresses can be accommodated by plastic deformation of ductile Al-coating. On the other hand, chromium is a hard and brittle transition metal, while CrN is a brittle ceramic material. Thus, both materials are more prone to fracture during heating and cooling. However, our previous work and current results showed that the adhesion between the Al and POM should be improved to increase the resistance to peeling. In our opinion, adhesion can be done by optimising the plasma surface activation step.

One-layer CrN coating adheres rather well to the POM substrate. It would be sensible to test the one-layer CrN coating with different thicknesses to find the optimal value regarding, e.g. scratch resistance and coefficient of friction. The application of several layers does not work well. The decohesion at the coating–substrate interface occurred by repeating heating and cooling, and with the increasing number of cycles, more and more coating is removed from the surface, thus decreasing the covering fraction.

The metallisation of POM by chromium appeared to be the most critical issue. Even if one-layer chromium coating was completely fractured, the application of the typical industrial parameters is inadequate. Since chromium was successfully metalised to different polymer substrates, it is possible that it can also be accomplished for POM. It is possible that the current Cr thickness lies above the optimum value. There are several parameters that can be varied by magnetron sputtering. The most important are sputtering power, substrate temperature, the minimum and maximum temperature during sputtering, sputtering rate, substrate rotation speed and plasma surface activation. It seems that also the flatness and roughness of POM substrate play an important role and that after maching, the surface should be made smoother. Thus the main goals in future investigations will be directed towards the improvement the adherence of different coating on POM substrate, optimising the thickness of a single layer and prevention of coating fracture during repeated metalisation.

Comment #3: The authors can add few sentences regarding te application part of these type of coatings which is more useful to the scientific community?

Response:  As explained in the Introduction of the revised manuscript, PVD coatings on the polymeric substrate are currently mainly used for optical components in the automotive industry. Furthermore, there are also several other fields where PVD coatings may be used: biomedical implants and other medical devices, the solar industry, cutting tools, etc. In all of these applications, the PVD processes can be mono-layered or multi-layered.

Round 2

Reviewer 1 Report

The authors replied to the referee’s comments and modified the manuscript according to the suggestions. Thus, I suggest accepting the paper.

Author Response

The authors replied to the referee’s comments and modified the manuscript according to the suggestions. Thus, I suggest accepting the paper.

Thank you.